# Creatine kinase is associated with glycated haemoglobin in a nondiabetic population. The Tromsø study

Svein Ivar Bekkelund[1,2]*

1 Department of Clinical Medicine, UiT – The Arctic University of Norway, Tromsø, Norway, 2 Department of Neurology, University Hospital of North Norway, Tromsø, Norway

* svein-ivar.bekkelund@uit.no

## Abstract

### Background

Creatine kinase (CK) has been associated with insulin resistance and identified as a risk marker of cardiovascular disease largely by its relationship with hypertension and increased body mass index. This study determined whether CK is a predictor of glycated haemoglobin (HbA$_{1C}$) in a nondiabetic general population.

### Methods

Associations between CK and the outcome variable HbA$_{1C}$ (%) were performed by variance and multivariate analyses in 11662 nondiabetic subjects defined as HbA$_{1C}$ (%) <6.5 who participated in the population based Tromsø study (Tromsø 6) in Norway.

### Results

Abnormal elevated CK was detected in 543/11662 participants (4.66%). Mean HbA$_{1C}$ (%) in the "high CK" group was 5.62 (SD = 0.33) compared to 5.52 (SD = 0.36) in the "normal CK" group, $P$ <0.001. CK increased significantly and linearly with higher levels of HbA$_{1C}$ (%) quartiles in women ($P$ <0.001) and non-linearly in men ($P$ <0.001). In a multivariate analysis, CK was independently associated with HbA$_{1C}$ (%) after adjusting for age, sex, body mass index, blood pressure, glucose, lipids, C-reactive protein, creatinine, alanine transaminase and aspartate aminotransferase. A 1-unit increase in log CK was associated with a 0.17-unit increase in HbA$_{1C}$ (%).

### Conclusion

These data demonstrate a positive and independent association between CK and glycated haemoglobin in a nondiabetic general population.

**Data Availability Statement:** Due to ethical and legal restrictions, data is only available upon request to the Tromsø Study. Any enquiries should be sent to the Institutional Data Access committee of The Tromsø study, Department of Community

Medicine, Faculty of Health Sciences, UiT- The Arctic University of Tromsø (tromsous@uit.no).

**Funding:** The author received no specific funding for this work.

**Competing interests:** The author has declared that no competing interests exist.

## Introduction

The enzymatic activity of creatine kinase (CK) is an important biological reaction in the formation of adenosine triphosphate (ATP), which is necessary for energy-demanding processes in human and animal cells, especially muscle contractions [1]. Both animal and human studies have shown relationships between CK and insulin resistance. Myocytes and adipocytes are important sites of insulin action, glucose uptake, and insulin resistance [2, 3], and ATP is involved in the muscular glucose uptake process [4]. Moreover, cytosolic CK reacts with glycolytic enzymes involved in ATP-generation, like pyruvate kinase [5].

Muscular activity, especially long-term and intense exercise and eccentric muscular training, may increase CK markedly [6, 7], while leisure physical exercise increases CK modestly; i.e., approximately 5% [8]. In addition to physiological elevation of CK, population studies have identified slightly increased CK as a possible cardiovascular disease (CVD) risk factor which include hypertension, obesity, and metabolic syndrome [9–12], although the mechanisms are largely unexplained. Thus, exercise increases insulin sensitivity and thereby reduces cardiovascular risk [13]. Differences in metabolic activity between muscle fibre types may explain some of the shared biological effects between glucose metabolism and CK. Higher CK activity and reduced insulin sensitivity are reported features of type 2B muscle fibres in contrast to type 1 fibres which to a greater extent promote oxidative metabolic reactions [14]. Animal studies have shown a shift from muscle fibre 2 to type 1 predominance upon CK inhibition and thereby stimulating oxidative phosphorylation, weight loss, and insulin sensitivity [15–17]. Consequently, subjects with relatively more 2B muscle fibres hypothetically run a higher CVD risk than those with muscle fibre type 1 predominance [18]. To further elaborate the connection between CK and glucose metabolism, this study hypothesized that CK is independently associated with glycated haemoglobin ($HbA_{1c}$) in a population of nondiabetic subjects recruited from a general Causation population.

## Materials and methods

The present cross-sectional designed study is based on data from the 6th Tromsø Study, which is a prospective population-based study recruiting inhabitants from the community of Tromsø, Norway. At the beginning in 1974, it mainly focused on cardiovascular diseases. Inhabitants of the municipality of Tromsø and samples from certain age groups of subjects previously participated in the survey (4th Tromsø study) plus a 10% random sample from age groups 30–39, all participants aged 40–49 and between 60–87 years (mean 58 years) were selected for inclusion. The data were collected from October 2007 to 19 September 2008. In total, 11662 mainly Causations (87.3% ethnic Norwegians, 1.6% Sami ethnicity, 1.3% Finnish origin, 2.2% of other ethnicities, and 7.6% without information about ethnicity) participated [19]. Those with diabetes (n = 1286) defined as $HbA_{1c} \geq 6.5\%$ were excluded. This definition was used since fasting blood sugar and use of antidiabetic medication were not measured in the Tromsø study. Written consent was obtained, and the Norwegian Committee for Medical and Health Research Ethics (REC) approved the study (reference number 121/2006).

All samples carried out in accordance with relevant guidelines and regulations are described elsewhere [20]. Serum-CK was obtained in an automated chemistry analyzer (Modular P, Roche) by photometry using an enzymatic method (CK-NAC, Roche Diagnostics, Mannheim, Germany) with an analytical variation coefficient ≤1.6%. The following CK reference intervals developed by the Nordic Reference Interval Project (NORIP) were used: Men 18–50 years: 50–400 U/L; men ≥50 years: 40–280 U/L and women: 35–210 U/L [21]. Subjects with CK ≥1000 U/L (7 men and 16 women) were regarded as outliers and excluded from the study. Serum alanine transaminase (ALT), aspartate aminotransferase (AST) and gamma

glutamyl transpeptidase (GGT) were analysed in an automated clinical chemistry analyzer (Modular P, Roche) by photometry, using an enzymatic method (CK-NAC, Roche Diagnostics, Mannheim, Germany).

Measurement of HbA1c in EDTA whole blood was based on an immunoturbidimetric assay (UNIMATES, F. Hoffmann-La Roche AG). The $HbA_{1c}$ (%) was calculated from the $HbA_{1c}$ /Hb ratio. Furthermore, non-fasting serum-glucose was obtained. Serum total cholesterol was analyzed by an enzymatic colorimetric method using a commercially available kit (CHOD-PAP, Boehringer-Mannheim, Mannheim, Germany). Serum high-density lipoprotein (HDL) cholesterol was measured after precipitation of lower-density lipoprotein (LDL) with heparin and manganese chloride. High-sensitivity C-reactive protein (hs-CRP) was analyzed using a particle-enhanced immunoturbidimetric assay on a Modular P (Roche Hitachi, Mannheim, Germany), with reagents from Roche Diagnostics (Mannheim, Germany). This was done in thawed aliquots after storage at −20˚C and analyzed in batches during the ongoing survey. The lower limit of the hs-CRP assay was detected at 0.03 mg /L-level and CRP lower <0.03 mg /L were set at this value. The analytical variation coefficient for hs-CRP levels between 0.1 mg /L and 20 mg /L was <4%.

All analyses were performed at the Department of Clinical Biochemistry, University Hospital of North Norway, Tromsø. According to the standard procedure in the Tromsø study, height and weight were measured wearing light clothing without shoes to the nearest 0.1 cm and 0.1 kg using an automatic device, and body mass index (BMI) was calculated as weight (kg) divided by height squared ($m^2$). Diabetes was defined as $HbA_{1c} \geq 6.5\%$. Use of lipid-lowering drugs (either currently, previous, or never use) and coronary heart disease (reported previous heart attack) were registered via standard questionnaires in the Tromsø study. Also, physical exercise assessed by intensity, frequency and duration were addressed via a self-administered questionnaire. Participants were asked about their physical activity in leisure time during the last year (weekly average for the year). Heavy exerciser was defined when subjects performed moderate or hard exercise ≥2 hours per week. High alcohol consumption was defined as drinking alcohol at least twice a week. Further details are published elsewhere [8]. An automatic device (Dinamap Vital Signs Monitor 1846; Critikon Inc, Tampa, FL) was used to record blood pressure. After 2 minutes rest in a sitting position, 3 readings were taken on the upper right arm at 1-minute intervals. Of them were the average of the 2 last readings used in the analyses. Hypertension was defined as systolic blood pressure ≥140 mmHg, diastolic blood pressure ≥90 mmHg, or use of antihypertensive medication.

## Statistical analysis

Study variables were evaluated by inspection of histograms, and calculation of kurtosis and skewness. The histograms showed right-sided skewness for CK (skewness 3.17, kurtosis 16.41), glucose (skewness 1.76, kurtosis 8.85), ALT (skewness 12.22, kurtosis 331.99), AST (skewness 14.13, kurtosis 355.59), and GGT (skewness 6.06, kurtosis 75.66). Inspection of histograms showed normal distribution of the variables after log-transformation and log-values were therefore used in the analyses (log CK; skewness 0.50, kurtosis 0.68). Descriptive data are presented as mean and standard deviation (SD) or number and frequency. Student´s t-test was used to calculate the differences between means and $\chi^2$- test to compare the frequencies of data. ANOVA was used to test differences between means of CK and $HbA_{1c}$ (%) quartiles, but also other relevant biomarkers/confounders and demographics. Confounding variables identified by comparisons between groups with high CK and normal CK and by variance analyses were used to model the association between CK and $HbA_{1c}$ (%). By multiple regression analysis, confounders with statistically significant values from these analyses

were tested and adjusted for in relation to $HbA_{1c}$ (%) as the dependent variable and log CK, age, BMI, systolic blood pressure, glucose, HDL- and LDL cholesterol, log hs-CRP, creatinine, ALT, and AST as independent variables. The data were reanalyzed in 9644 subjects after excluding those who reported moderate or hard leisure physical exercise ≥2 hours per week (n = 2018) and secondly after excluding1693 lipid lowering drug users (n = 7951). Two-sided $P$ <0.05 was considered statistically significant. All analyses were conducted by SPSS software (Statistical Package for Social Science INC, Chicago, Illinois, USA), version 26.

## Results

Mean CK was 121,71 (SD = 83.03) U/L [men: 148.65 U/L (SD = 96.79), women: 98.56 (SD = 60.07) U/L]. CK values above upper normal limit was detected in 543/11662 (4.66%) participants; 289/5368 (5.38%) men and 254/6294 (4.04%) women. This subgroup had higher BMI, systolic and diastolic blood pressure, creatinine, ALT, and AST than the others (Table 1). Furthermore, $HbA_{1C}$ (%) and blood glucose were both increased in the "high CK" group compared to the "normal CK" group (Table 1). High CK and covariates significantly associated with increased $HbA_{1C}$ (%) either by direct comparisons (Table 1) or by variance analysis (Table 2) were included in subsequent multivariate analyses (Table 3).

**Table 1. Clinical characteristics of the study participants with and without hyperCKemia.** Data are presented as mean (SD) or numbers (%).

| Variables | High CK* (n = 543) | Normal CK (n = 11119) | P-value |
|---|---|---|---|
| Age (years) | 56.96 (12.51) | 61.87 (9.66) | <0.001 |
| Men | 289 (53.22) | 5079 (45.68) | <0.001 |
| BMI (kg/m²) | 27.23 (4.07) | 26.65 (4.16) | 0.002 |
| Obesity (BMI ≥30 kg/m²) | 125 (23.02) | 2045 (18.39) | 0.005 |
| Heavy exerciser* | 113 (20.81) | 1905 (17.13) | 0.018 |
| High alcohol consumption | 122 (22.47) | 2495 (22.44) | 0.961 |
| S-total cholesterol (mmol/l) | 5.70 (1.03) | 5.64 (1.09) | 0.194 |
| HDL cholesterol (mmol/l) | 1.56 (0.44) | 1.53 (0.44) | 0.127 |
| LDL cholesterol (mmol/l) | 3.63 (0.91) | 3.58 (0.95) | 0.195 |
| Use of lipid lowering drugs | 62 (11.41) | 1631 (14.66) | 0.79 |
| Systolic BP (mm Hg) | 139.27 (21.1) | 134.80 (23.05) | <0.001 |
| Diastolic BP (mm Hg) | 79.34 (9.94) | 77.72 (10.71) | <0.001 |
| Hypertension | 117 (21.55) | 2515 (22.62) | 0.633 |
| Coronary heart disease | 26 (4.79) | 494 (4.44) | 0.677 |
| Hs-CRP (mg/dl)** | 2.12 (2.63) | 2.43 (4.54) | 0.538 |
| S-creatinine (μmol/l) | 73.06 (15.51) | 69.77 (15.72) | <0.001 |
| S-ALT (U/L)** | 31.83 (14.60) | 28.49 (21.98) | <0.001 |
| S-AST (U/L)** | 32.51 (8.50) | 30.79 (31.23) | <0.001 |
| S-GOT (U/L)** | 28.19 (19.04) | 30.79 (31.23) | 0.829 |
| S-glucose (mmol/l)** | 5.13 (0.59) | 5.07 (0.63) | 0.004 |
| S-HbA1c (%) | 5.62 (0.33) | 5.52 (0.36) | <0.001 |
| S-CK (U/L)** | 110.79 (58.83) | 337.82 (158.61) | <0.001 |

ALT, alanine transaminase; AST, aspartate aminotransferase; BMI, body mass index; CK, creatine kinase; BP, blood pressure; hs-CRP, high-sensitive C-reactive protein; HbA1c, glycated haemoglobin; GGT, gamma glutamyl transpeptidase; HDL, high-density lipoprotein; LDL, low-density lipoprotein; *Moderate or hard leisure physical exercise ≥2times per week; *Reference limits for normal CK: Men < 50 years: 50–400 U/L, men ≥ 50 years: 40–280 U/L, women: 35–210 U/L; **Calculated log-transformed

**Table 2. Clinical characteristics and endpoint variables in 11662 participants by HbA$_{1C}$ (%) quartiles.** Data are presented as mean (SD) or numbers (%).

| Variables | HbA$_{1C}$ (%) intervals | | | | |
|---|---|---|---|---|---|
| | <5.300 | 5.300–5.499 | 5.500–5.799 | ≥5.800 | P-value for trend |
| N | 2604 | 2372 | 3590 | 3096 | |
| Log CK (U/L), men | 2.09 (0.22) | 2.11 (0.24) | 2.12 (0.24) | 2.10 (0.23) | <0.001 |
| Log CK (U/L), women | 1.91 (0.20) | 1.93 (0.20) | 1.96 (0.20) | 1.96 (0.21) | <0.001 |
| High CK | 69 (2.65) | 106 (4.47) | 189 (5.25) | 177 (5.71) | <0.001 |
| Men | 1039 (39.90) | 1076 (45.36) | 1809 (50.26) | 1545 (49.90) | <0.001 |
| Age (years) | 49.64 (11.54) | 54.39 (11.99) | 58.45 (11.77) | 63.49 (10.93) | <0.001 |
| BMI (kg/m$^2$) | 25.66 (3.82) | 26.34 (3.93) | 26.73 (4.05) | 27.78 (4.42) | <0.001 |
| Log s-glucose (mmol/l) | 0.68 (0.05) | 0.69 (0.05) | 0.70 (0.05) | 0.72 (0.06) | <0.001 |
| Total cholesterol (mmol/L) | 5.28 (1.00) | 5.59 (1.03) | 5.77 (1.05) | 5.82 (1.36) | <0.001 |
| HDL cholesterol (mmol/L) | 1.57 (0.44) | 1.54 (0.44) | 1.54 (0.44) | 1.47 (0.42) | <0.001 |
| LDL cholesterol (mmol/L) | 3.26 (0.87) | 3.54 (0.89) | 3.68 (0.92) | 3.74 (1.01) | <0.001 |
| Log CRP (mg/L) | 0.01 (0.42) | 0.07 (0.42) | 0.15 (0.42) | 0.25 (0.42) | <0.001 |
| S-creatinine (µmol/l) | 67.61 (13.01) | 69.63 (14.72) | 70.59 (15.66) | 71.58 (18.11) | <0.001 |
| Log ALT (U/L | 1.36 (0.22) | 1.37 (0.21) | 1.40 (0.20) | 1.43 (0.21) | <0.001 |
| Log AST (U/L | 1.37 (0.12) | 1.39 (0.12 | 1.40 (0.13) | 1.42 (0.13) | <0.001 |

ALT, alanine transaminase; AST, aspartate aminotransferase; BMI, body mass index; CK, creatine kinase; hs-CRP, high-sensitive C-reactive protein; HbA1c, glycated haemoglobin; HDL, high-density lipoprotein; LDL, low-density lipoprotein

Table 2 compares log CK with HbA$_{1C}$ (%) quartiles. The main results were a significant linear association between higher log-CK levels and increasing HbA$_{1C}$ (%) in women and a similar significant, but nonlinear relationship in men. Table 2 also shows the relationships for confounders. All variables positively associated with HbA$_{1C}$ (%) in the variance analyses were

**Table 3. Multiple regression analyses comparing glycated haemoglobin (dependent variable) with independent variables.**

| Independent variables | HbA$_{1C}$ (%) as dependent variable | | | | | |
|---|---|---|---|---|---|---|
| | Model 1 | | | Model 2 | | |
| | ß* (SE) | 95% CI | P-value | ß* (SE) | 95% CI | P-value |
| Log CK (U/L) | 0.172 (0.023) | 0.127 to 0.217 | <0.001 | 0.164 (0.025) | 0.115 to 0.214 | <0.001 |
| Age (years) | 0.011 (0.000) | 0.010 to 0.012 | <0.001 | 0.011 (0.000) | 0.010 to 0.012 | <0.001 |
| Sex | -0.030 (0.007) | -0.053 to -0.007 | 0.010 | -0.027 (0.013) | -0.052 to 0.002 | 0.033 |
| BMI (kg/m$^2$) | 0.005 (0.001) | 0.003 to 0.008 | <0.001 | 0.005 (0.001) | 0.003 to 0.008 | <0.001 |
| Log s-glucose (mmol/l) | 1.504 (0.098) | 1.313 to 1.695 | <0.001 | 1.517 (0.107) | 1.307 to 1.728 | <0.001 |
| HDL cholesterol (mmol/L) | -0.070 (0.012) | -0.094 to—0.047 | <0.001 | -0.076 (0.013) | -0.102 to -0.050 | <0.001 |
| LDL cholesterol (mmol/L) | 0.032 (0.005) | 0.022 to 0.041 | <0.001 | 0.029 (0.005) | 0.018 to 0.039 | <0.001 |
| Log CRP (mg/L) | 0.090 (0.011) | 0.068 to 0.112 | <0.001 | 0.088 (0.012) | 0.064 to 0.113 | <0.001 |
| S-creatinine (µmol/l) | 0.000 (0.000) | -0.001 to 0.000 | 0.602 | 0.000 (0.000) | -0.001 to 0.000 | 0.363 |
| Systolic BP (mm Hg) | -0.001 (0.000) | -0.001 to 0.000 | <0.001 | -0.001 (0.000) | -0.001 to 0.000 | 0.002 |
| Log ALT (U/L | 0.115 (0.035) | 0.045 to 0.184 | 0.001 | 0.115 (0.039) | 0.038 to 0.191 | 0.004 |
| Log AST (U/L | -0.112 (0.061) | -0.232 to 0.007 | 0.064 | -0.140 (0.087) | -0.271 to -0.009 | 0.036 |
| Adjusted R$^2$ | 0.25 | | | 0.25 | | |

ALT, alanine transaminase; AST, aspartate aminotransferase; BMI, body mass index; BP, blood pressure; CK, creatine kinase; HDL, high-density lipoprotein; LDL, low-density lipoprotein; hs-CRP, high-sensitive C-reactive protein. Model 1, total sample; model 2, total sample excluding heavy physical exercisers;
*Values are regression coefficients (95% CI) expressed in HbA$_{1C}$ (%) for a 1-unit change in independent variables.

included as independent variables in the subsequent multivariate analysis (Table 3). Furthermore, univariate association between log CK and HbA$_{1C}$ (%) was statistically significant (ß = 0.055, 95% CI: 0.044 to 0.067, $P$ <0.001). A similar relationship was found for log glucose vs. log CK (ß = 0.293, 95% CI: 0.212 to 0.375, $P$ <0.001). Log CK was independently associated with HbA$_{1C}$ (%) when adjusted for age, sex, BMI, lipids, log glucose, systolic blood pressure, hs-CRP, creatinine, ALT, and AST in a multiple regression model (Table 3). A 1-unit increase in log CK was associated with 0.17-unit increase in HbA$_{1C}$ (%) (Table 3). These determinants explained 25% of the variance in HbA$_{1C}$ (Table 3). Sex-stratified analyses showed independent relationships for both men (ß = 0.008, 95% CI: 0.005 to 0.011, $P$ <0.001) and women (ß = 0.011, 95% CI: 0.00 to 0.014, $P$ <0.001). Reanalyses of the data after removing 2018 heavy exercisers confirmed CK as an independent predictor of HbA$_{1C}$ (%) (Table 3). When additionally removing 1693 subjects using lipid lowering drugs, CK predicted HbA$_{1C}$ (%) as followed: ß = 0.154, 95% CI: 0.098 to 0.205, $P$ <0.001 (n = 7951).

## Discussion

The previously proposed relationship between CK and impaired glucose metabolism is confirmed and further developed in the present study by demonstrating a significant positive and independent association between CK and HbA$_{1C}$ (%).

Subclinical CK elevation is reported in diabetic subjects [22], and has been attributed to the presence of metabolic syndrome [9]. In a cohort of 92 diabetics, 19 (21%) had elevated CK-levels. Of them were 7 suspected to have metabolic myopathy [22]. Thus, subjects with ≥2 metabolic syndrome features had higher CK than others, indicating a synergetic effect [9]. Moreover, the myocardial CK isotype (CK-MB) was increased in diabetic patients with additional metabolic syndrome [23] and was linearly related to HbA$_{1C}$ (%) in type 2 diabetes mellitus [23]. HbA$_{1C}$ (%), but not total CK, was associated with major cardiac events in a retrospective study group of nondiabetic patients with initial nonfatal myocardial infarction [24]. Despite indirect evidence for a relationship between CK and impaired glucose metabolism, also supported by the present study, the associations between HbA$_{1C}$ (%) and CK have not been subject to primary investigation in population-based studies before. In an Asian population study, among 4562 individuals including 373 (8.2%) with diabetes mellitus, CK were positively associated with CVD risk factors such as blood pressure and BMI, but negatively associated with HbA$_{1C}$ (%) and fasting blood sugar in contrast to the present study [25]. The CK- HbA$_{1C}$ (%) relationship was not the primary target of investigation in that study however, and no effect of covariates were estimated. Furthermore, CK vs. fasting glucose data and the number/frequencies of diabetes mellitus cases distributed along quartiles of CK intervals were nonlinear [25].

HbA$_{1C}$ (%) and non-fasting blood sugar increased clearly with age in the present study. A Finnish population study reported HbA$_{1C}$ (%) to be primarily associated with fasting glucose and CRP (10%) and to a lesser degree with insulin sensitivity and insulin secretion (2%) in 9398 nondiabetic men [26]. In addition, the independent relationship between HbA$_{1C}$ (%) and unfavourable lipid profile in the present study is discussed with different outcomes in the literature [27, 28]. Whether HbA$_{1C}$ (%) is more or less related to nonglycemic factors is unclarified.

An important limitation to the study is lack of causal information from the analyses of the HbA (%) vs. CK relationship due to the cross-sectional design. Possible influence of confounding factors is therefore important to be aware of. Muscle fibre 2B preponderance indicating impaired insulin sensitivity is associated with increased BMI [14, 29]. Moreover, higher frequencies of muscle fibre type 2B seem to increase with age, hypothetically supporting the view

that age may confound the association between CK and HbA$_{1C}$ (Table 2) [30]. Thus, muscle fibre morphology may therefore explain why CK associates with HbA$_{1C}$ (%). Age and BMI appear as confounders since both were positively and independently associated with HbA$_{1C}$ (%) and CK in the present study. Similar relationships between age, BMI and CK are reported by others [10, 12, 31] but less is known about the role of HbA$_{1C}$ (%). This also include sex-differences in this relationship as CK associated linearly with HbA$_{1C}$ (%) in women but not in men. Physical activity is an obvious confounder in such a study. Despite reanalyzing the data after excluding heavy physical exercisers, CK remained an independent predictor of HbA$_{1C}$ (%). The impact of leisure physical activity on the CK-values in this population is previously estimated to be about 5% [8]. Likewise, a positive association between BMI, blood pressure and CK and an inverse relationship between CK and CRP are also previously reported [11, 20]. In line with findings from others, this study did not show higher CK in statin users [10]. Neither is ethnicity expected to play a major role since Caucasian populations like the present one is associated with lower CK levels than black- or mixed ethnic populations [10, 12], Knowledge about confounding variables related to CK may therefore be regarded as a strength of this study.

## Conclusion

In summary, creatine kinase was associated with glycated haemoglobin in this Norwegian population-based study. Although CK is not proven to be an etiologic factor in the CVD risk panorama, such a view is supported by its connection with HbA$_{1C}$ (%) in this study. This finding needs confirmation and to be verified in prospective studies.

## Acknowledgments

I am indebted to the Norwegian Institute of Public Health for organizing data collection in the sixth survey of the Tromsø Study.

## Author Contributions

**Conceptualization:** Svein Ivar Bekkelund.

**Data curation:** Svein Ivar Bekkelund.

**Formal analysis:** Svein Ivar Bekkelund.

**Investigation:** Svein Ivar Bekkelund.

**Methodology:** Svein Ivar Bekkelund.

**Project administration:** Svein Ivar Bekkelund.

**Software:** Svein Ivar Bekkelund.

**Validation:** Svein Ivar Bekkelund.

**Writing – original draft:** Svein Ivar Bekkelund.

**Writing – review & editing:** Svein Ivar Bekkelund.

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
