## [Decision Letter · Decision Letter 0]

20 Jun 2022

PONE-D-21-32482

Creatine kinase is associated with glycated haemoglobin in a nondiabetic population. The Tromsø study

PLOS ONE

Dear Dr. Bekkelund,

Thank you for submitting your manuscript to PLOS ONE. After careful consideration, we feel that it has merit but does not fully meet PLOS ONE’s publication criteria as it currently stands. Therefore, we invite you to submit a revised version of the manuscript that addresses the points raised during the review process.

Please address all reviewers' comments and revise your manuscript accordingly. The major factor that may jeopardize the validity of the study is lack of physical activity as a confounding factor. As you mentioned in both introduction and discussion, physical activity and muscle types may related to CK concentrations.

We look forward to receiving your revised manuscript.

Kind regards,

Shengxu Li, MD, MPH, PhD

Academic Editor

PLOS ONE

Journal Requirements:

2. Thank you for stating the following in the Acknowledgments Section of your manuscript: "I am indebted to the Norwegian Institute of Public Health for their participation in the data collection in the sixth survey of the Tromsø Study. The publication charges for this article have been funded by a grant from the publication fund of UiT - The Arctic University of Norway. "

Please remove any funding-related text from the manuscript and let us know how you would like to update your Funding Statement. Currently, your Funding Statement reads as follows: "The author received no specific funding for this work"

Additional Editor Comments (if provided):

Please respond to the reviewers' comments, in particular comments on factors that may influence CK concentrations.

Reviewers' comments:

Reviewer's Responses to Questions

**Comments to the Author**

1. Is the manuscript technically sound, and do the data support the conclusions?

Reviewer #1: Yes

Reviewer #2: Partly

2. Has the statistical analysis been performed appropriately and rigorously? 

Reviewer #1: Yes

Reviewer #2: Yes

3. Have the authors made all data underlying the findings in their manuscript fully available?

Reviewer #1: Yes

Reviewer #2: No

4. Is the manuscript presented in an intelligible fashion and written in standard English?

Reviewer #1: Yes

Reviewer #2: No

5. Review Comments to the Author

Reviewer #1: Manuscript #: PONE-D-21-32482

Title: Creatine kinase is associated with glycated haemoglobin in a nondiabetic population. The Tromsø study

CK is mainly influenced by the degree of physical activity, the type and number of drugs , and genetic factors. All this variables were not included in the present study. The influence of the Hba1c on CK can be only reliably assessed if the cohort had been homogenised for these inlfuences

1/22

Reviewer #2: The study examined the associations between CK and other CV risk factors in a large sample. It was found that CK was associated with these factors. These associations, however, were cross-sectional and important factors, such as physical activity, were not considered.

1) As this is a cross-sectional study, causal relationships cannot be established.

2) As the authors discussed, an important factor influencing CK concentrations is physical activity, which was lacking in the study.

3) Systolic blood pressure is closed correlated to diastolic blood pressure. Including these two in the same model might have resulted colinearity. It would be advisable to use separate models for blood pressure.

4) According to the authors, the reference values for CK were age-dependent for men, but not for women (ref 21). Could the authors explain why?

5) Language in the manuscript needs further clean-up.

6. PLOS authors have the option to publish the peer review history of their article (what does this mean?). If published, this will include your full peer review and any attached files.

Reviewer #1: No

Reviewer #2: No

---

## [Author Response · Author response to Decision Letter 0]

29 Jul 2022

Editor, PLOS ONE

Thank you for reviewing the manuscript and for considering it for publication in PLOS ONE. The responses to reviewers are carefully considered and described point by point below. 

Reviewer #1: Manuscript #: PONE-D-21-32482

Title: Creatine kinase is associated with glycated haemoglobin in a nondiabetic population. The Tromsø study

CK is mainly influenced by the degree of physical activity, the type and number of drugs, and genetic factors. All these variables were not included in the present study. The influence of the Hba1c on CK can be only reliably assessed if the cohort had been homogenised for these inlfuences

Reply:

Physical activity has been included in the variable list (Table 1) and adjusted for in the multiple variance analysis (Table 4). The main finding of the study; an independent association between glycated haemoglobin and creatin kinase, remained unchanged after adjusting for physical exercise. Definition of “heavy exerciser” is described in the methods.

Reviewer #2: The study examined the associations between CK and other CV risk factors in a large sample. It was found that CK was associated with these factors. These associations, however, were cross-sectional and important factors, such as physical activity, were not considered.

Reply: See above.

1) As this is a cross-sectional study, causal relationships cannot be established.

Reply: See below

2) As the authors discussed, an important factor influencing CK concentrations is physical activity, which was lacking in the study.

3) Systolic blood pressure is closed correlated to diastolic blood pressure. Including these two in the same model might have resulted colinearity. It would be advisable to use separate models for blood pressure.

Reply:

The multivariate analyses (Tables 3 and 4) are recalculated by including systolic blood pressure (model 1) and diastolic blood pressure (model 2) separately. These two models are presented side by side in the tables and explained in “Materials and methods” (statistical analyses).

4) According to the authors, the reference values for CK were age-dependent for men, but not for women (ref 21). Could the authors explain why?

5) Language in the manuscript needs further clean-up.

Reply:

I have reviewed the manuscript and tried to improve the language. Especially, the last paragraph in the discussion and the conclusion are modified. Limitations to the study and comments upon sex-differences are also included:

“An important limitation to the study is lack of causal information from the analyses of the HbA1C vs. CK relationship due to the cross-sectional design. Possible influence of confounding factors is therefore important to be aware of. Muscle fibre 2B preponderance indicating impaired insulin sensitivity is associated with increased BMI [14, 29]. Moreover, higher frequencies of muscle fibre type 2B seem to increase with age, supporting the view that age may confound the association between HbA1C and CK (Table 2) [30]. Hypothetically, muscle fibre morphology may therefore explain why CK associates with HbA1C. Age and BMI appear as confounders since both were positively and independently associated with HbA1C and CK in the present study. Similar relationships between age, BMI and CK are reported by others [10, 12, 31] but less is known about the role of HbA1C. This also include sex-differences in this relationship as CK associated linearly with HbA1C in women but not in men”.

Response to the Editor:

The funding paragraph is changed to: “The author received no specific funding for this work”.

Best regards,

Svein Ivar Bekkelund 

Corresponding author

---

## [Decision Letter · Decision Letter 1]

17 Oct 2022

PONE-D-21-32482R1Creatine kinase is associated with glycated haemoglobin in a nondiabetic population. The Tromsø studyPLOS ONE

Dear Dr. Bekkelund,

Thank you for submitting your manuscript to PLOS ONE. One of the original reviewers continues to have major concerns. To give your revised manuscript a more thorough review, I invited another reviewer with appropriate expertise to review your manuscript. This reviewer raised additional concerns. Therefore, we invite you to submit a revised version of the manuscript that addresses the points raised during the review process. Please submit your revised manuscript by Dec 01 2022 11:59PM. If you will need more time than this to complete your revisions, please reply to this message or contact the journal office at plosone@plos.org. Please include the following items when submitting your revised manuscript:A rebuttal letter that responds to each point raised by the academic editor and reviewer(s). You should upload this letter as a separate file labeled 'Response to Reviewers'.A marked-up copy of your manuscript that highlights changes made to the original version. You should upload this as a separate file labeled 'Revised Manuscript with Track Changes'.An unmarked version of your revised paper without tracked changes. You should upload this as a separate file labeled 'Manuscript'.

We look forward to receiving your revised manuscript.

Kind regards,

Shengxu Li, MD, MPH, PhD

Academic Editor

PLOS ONE

Reviewers' comments:

Reviewer's Responses to Questions

**Comments to the Author**

1. If the authors have adequately addressed your comments raised in a previous round of review and you feel that this manuscript is now acceptable for publication, you may indicate that here to bypass the “Comments to the Author” section, enter your conflict of interest statement in the “Confidential to Editor” section, and submit your "Accept" recommendation.

Reviewer #1: All comments have been addressed

Reviewer #3: (No Response)

2. Is the manuscript technically sound, and do the data support the conclusions?

Reviewer #1: Partly

Reviewer #3: No

3. Has the statistical analysis been performed appropriately and rigorously? 

Reviewer #1: Yes

Reviewer #3: No

4. Have the authors made all data underlying the findings in their manuscript fully available?

Reviewer #1: Yes

Reviewer #3: Yes

5. Is the manuscript presented in an intelligible fashion and written in standard English?

Reviewer #1: Yes

Reviewer #3: Yes

6. Review Comments to the Author

Reviewer #1: Manuscript #: PONE-D-21-32482R1

Title: Creatine kinase is associated with glycated haemoglobin in a nondiabetic population. The Tromsø study

The result and the main message of the study have not significantly changed. Other factors determining CK were not sufficiently included in the evlautaion.

8/22

Reviewer #3: This study aims to determine the association between creatine kinase and glycated haemoglobin (HbA1C) in a nondiabetic population. The authors found that creatine kinase (CK) was significantly and positively correlated with HbA1C in nondiabetic participants of the Tromsø Study. The sample size was large with a sufficient statistical power, and the finding is of interest. However, there are problems and concerns that need be addressed as below.

The major concern is that it is confusing throughout the paper regarding which one, HbA1c or CK, was used as the outcome variable.

In the Abstract, “Endpoint variables were measured cross-sectionally in 11662 nondiabetic subjects defined as HbA1C (%) <6.5 participating in the population-based Tromsø study (Tromsø 6) in Norway.” This statement is rather vague. It is not clear which variable was used as the outcome. “Additionally, HbA1C (%) was an independent predictor of high CK” indicated that CK is the outcome variable.

The conclusion “These data confirm a relationship between CK and glycosylation and conforms with the knowledge of the role of CK as a cardiovascular risk marker.” is not appropriate and needs to be revised. The data do not support “the role of CK as a cardiovascular risk marker”.

In Table 3, HbA1c was the dependent variable, but CK was the dependent variable in Table 4.

Diabetes was defined as HbA1c ≥6.5%. Please justify why glucose and anti-diabetic medication were not used for diagnosis.

Systolic and diastolic blood pressure should not be included in the same model. Their association parameters in Tables 3 and 4 are hard to explain due to the collinearity. Inclusion of systolic blood pressure is good enough.

7. PLOS authors have the option to publish the peer review history of their article (what does this mean?). If published, this will include your full peer review and any attached files.

Reviewer #1: No

Reviewer #3: No

---

## [Author Response · Author response to Decision Letter 1]

22 Nov 2022

Editor, PLoS ONE

PONE-D-21-32482R1

“Creatine kinase is associated with glycated haemoglobin in a nondiabetic population. The Tromsø study”.

The responses to reviewers 1 and 3 and from the Editor are carefully considered and described point by point below. 

Reviewer 1

The result and the main message of the study have not significantly changed. Other factors determining CK were not sufficiently included in the evaluation.

Reply:

Alcohol consumption, alanine transaminase, aspartate aminotransferase and gamma glutamyl transpeptidase are included, in Table 1 and those which were statically significantly associated with CK were used in the further analyses. The method of recruiting confounders is described in “statistical analyses”:

Confounding variables identified by comparisons between groups with high CK and normal CK and by variance analyses were used to model the association between CK and HbA1c (%). By multiple regression analysis, confounders with statistically significant values from these analyses were tested and adjusted for in relation to HbA1c (%) as the dependent variable and log CK, age, BMI, systolic blood pressure, glucose, HDL- and LDL cholesterol, log hs-CRP, creatinine, ALT, and AST as independent variables. The data were reanalyzed in 9644 subjects after excluding those who reported moderate or hard leisure physical exercise ≥2 hours per week (n=2018) and secondly after excluding1693 lipid lowering drug users (n=7951).

A discussion of CK-related confounders is included at the end of the discussion: 

Physical activity is an obvious confounder in such a study. Despite reanalysing the data after excluding heavy physical exercisers, CK remained an independent predictor of HbA1C (%). The impact of leisure physical activity on the CK-values in this population is previously estimated to be about 5% [8]. Likewise, a positive association between BMI, blood pressure and CK and an inverse relationship between CK and CRP are also previously reported [11, 20]. In line with findings from others, this study did not show higher CK in statin users [10]. Neither is ethnicity expected to play a major role since Caucasian populations like the present one is associated with lower CK levels than black- or mixed ethnic populations [10, 12], Knowledge about confounding variables related to CK may therefore be regarded as a strength of this study.

Reviewer 3

The major concern is that it is confusing throughout the paper regarding which one, HbA1c or CK, was used as the outcome variable. 

Reply:

The article is reviewed and the outcome variable HbA1C and CK as the independent variable are made clear.

In the Abstract, “Endpoint variables were measured cross-sectionally in 11662 nondiabetic subjects defined as HbA1C (%) <6.5 participating in the population-based Tromsø study (Tromsø 6) in Norway.” This statement is rather vague. It is not clear which variable was used as the outcome. “Additionally, HbA1C (%) was an independent predictor of high CK” indicated that CK is the outcome variable. 

Reply:

The method part in the abstract is changed to:

Associations between CK and the outcome variable HbA1C (%) were performed by variance and multivariate analyses in 11662 nondiabetic subjects defined as HbA1C (%) <6.5 who participated in the population based Tromsø study (Tromsø 6) in Norway.

The conclusion “These data confirm a relationship between CK and glycosylation and conforms with the knowledge of the role of CK as a cardiovascular risk marker.” is not appropriate and needs to be revised. The data do not support “the role of CK as a cardiovascular risk marker”.

Reply:

The conclusion is changed to:

These data demonstrate a positive and independent association between CK and glycated haemoglobin in a nondiabetic general population.

In Table 3, HbA1c was the dependent variable, but CK was the dependent variable in Table 4.

Reply:

Table 4 is deleted. Table 3 is now divided into models 1 (total sample) and 2 (total sample excluding heavy exercisers). Exercise as a confounder is commented upon in the last part of the discussion as referred to above (response to reviewer 1).

Accordingly, the background paragraph is changed to:

..This study determined whether CK is a predictor of glycated haemoglobin (HbA1C) in a nondiabetic general population. 

The last sentence in the Introduction is modified:

To further elaborate the connection between CK and glucose metabolism, this study hypothesized that CK is independently associated with glycated haemoglobin (HbA1c) in a population of nondiabetic subjects recruited from a general Causation population.

Diabetes was defined as HbA1c ≥6.5%. Please justify why glucose and anti-diabetic medication were not used for diagnosis. 

Reply:

This is explained in the methods:

Those with diabetes (n = 1286) defined as HbA1c ≥6.5% were excluded. This definition was used since fasting blood sugar and use of antidiabetic medication were not measured in the Tromsø study.

Systolic and diastolic blood pressure should not be included in the same model. Their association parameters in Tables 3 and 4 are hard to explain due to the collinearity. Inclusion of systolic blood pressure is good enough.

Reply:

Diastolic blood pressure is taken out of the model in Table 3. Table 4 is deleted.

---

## [Decision Letter · Decision Letter 2]

19 Jan 2023

Creatine kinase is associated with glycated haemoglobin in a nondiabetic population. The Tromsø study

PONE-D-21-32482R2

Dear Dr. Bekkelund

We’re pleased to inform you that your manuscript has been judged scientifically suitable for publication and will be formally accepted for publication once it meets all outstanding technical requirements.

Kind regards,

Shengxu Li, MD, MPH, PhD

Academic Editor

PLOS ONE

Additional Editor Comments (optional):

Reviewers' comments:

Reviewer's Responses to Questions

**Comments to the Author**

1. If the authors have adequately addressed your comments raised in a previous round of review and you feel that this manuscript is now acceptable for publication, you may indicate that here to bypass the “Comments to the Author” section, enter your conflict of interest statement in the “Confidential to Editor” section, and submit your "Accept" recommendation.

Reviewer #1: (No Response)

Reviewer #3: All comments have been addressed

2. Is the manuscript technically sound, and do the data support the conclusions?

Reviewer #1: Yes

Reviewer #3: Yes

3. Has the statistical analysis been performed appropriately and rigorously? 

Reviewer #1: Yes

Reviewer #3: Yes

4. Have the authors made all data underlying the findings in their manuscript fully available?

Reviewer #1: Yes

Reviewer #3: Yes

5. Is the manuscript presented in an intelligible fashion and written in standard English?

Reviewer #1: Yes

Reviewer #3: Yes

6. Review Comments to the Author

Reviewer #1: Manuscript #: PONE-D-21-32482R2

Title: Creatine kinase is associated with glycated haemoglobin in a nondiabetic population. The Tromsø study

The result and the main message of the study have not significantly changed. Factors such as medication, previous disease, family history, physical activity, diet were not considered.

11/22

Reviewer #3: Spell out HbA1c at its first use in the Abstract.

Change “multivariate analyses” to “multivariable analyses”.

Correct the typo “a general Causation population”.

7. PLOS authors have the option to publish the peer review history of their article (what does this mean?). If published, this will include your full peer review and any attached files.

Reviewer #1: No

Reviewer #3: No

---

## [Editor Report · Acceptance letter]

24 Jan 2023

PONE-D-21-32482R2 

Creatine kinase is associated with glycated haemoglobin in a nondiabetic population. The Tromsø study 

Dear Dr. Bekkelund:

I'm pleased to inform you that your manuscript has been deemed suitable for publication in PLOS ONE. Congratulations! Your manuscript is now with our production department. 

Kind regards, 

on behalf of

Dr. Shengxu Li 

%CORR_ED_EDITOR_ROLE%

PLOS ONE